# The transcriptional cofactor TRIM33 prevents apoptosis in B lymphoblastic leukemia by deactivating a single enhancer

Eric Wang[1], Shinpei Kawaoka[1†], Jae-Seok Roe[1], Junwei Shi[1,2], Anja F Hohmann[1], Yali Xu[1,2], Anand S Bhagwat[1,3], Yutaka Suzuki[4], Justin B Kinney[1,2], Christopher R Vakoc[1]*

[1]Cold Spring Harbor Laboratory, New York, United States; [2]Molecular and Cellular Biology Program, Stony Brook University, New York, United States; [3]Medical Scientist Training Program, Stony Brook University, New York, United States; [4]Department of Medical Genome Sciences, University of Tokyo, Kashiwa, Japan

**Abstract** Most mammalian transcription factors (TFs) and cofactors occupy thousands of genomic sites and modulate the expression of large gene networks to implement their biological functions. In this study, we describe an exception to this paradigm. TRIM33 is identified here as a lineage dependency in B cell neoplasms and is shown to perform this essential function by associating with a single *cis* element. ChIP-seq analysis of TRIM33 in murine B cell leukemia revealed a preferential association with two lineage-specific enhancers that harbor an exceptional density of motifs recognized by the PU.1 TF. TRIM33 is recruited to these elements by PU.1, yet acts to antagonize PU.1 function. One of the PU.1/TRIM33 co-occupied enhancers is upstream of the pro-apoptotic gene *Bim*, and deleting this enhancer renders TRIM33 dispensable for leukemia cell survival. These findings reveal an essential role for TRIM33 in preventing apoptosis in B lymphoblastic leukemia by interfering with enhancer-mediated *Bim* activation.

*For correspondence: vakoc@cshl.edu

Present address: †The Thomas N Sato BioMEC-X Laboratories, Advanced Telecommunications Research Institute International, Kyoto, Japan

Competing interests: The authors declare that no competing interests exist.

## Introduction

DNA-binding transcription factors (TFs) typically recognize short 6–12 base-pair motifs, which exist at a high frequency in mammalian genomes (*Wunderlich and Mirny, 2009*; *Spitz and Furlong, 2012*). While chromatin restricts the accessibility of these sequences, TFs nonetheless occupy thousands of genomic sites in any given cell type (*Spitz and Furlong, 2012*). As examples, MyoD has been detected at ~39,000 sites in primary myotubes and Pax5 at ~20,000 sites in B cell progenitors (*Cao et al., 2010*; *Revilla et al., 2012*). Furthermore, transcriptional cofactors (coactivators and corepressors) are also known to occupy the mammalian genome in a generalized manner (*Ram et al., 2011*). Thus, strong support exists for a model in mammalian cells in which transcriptional control is performed by regulators that exert broad, yet highly coordinated effects on gene expression (*Wunderlich and Mirny, 2009*; *Spitz and Furlong, 2012*). This is in stark contrast to prokaryotic species, in which TFs can occupy as few as one or two sites in an entire genome to implement precise transcriptional responses (*Martínez-Antonio and Collado-Vides, 2003*; *Wunderlich and Mirny, 2009*). Whether such a degree of gene-specificity exists among mammalian transcriptional regulators is currently unknown.

TRIM33, also known as TIF1γ, belongs to the tripartite motif (TRIM)-containing protein family of E3 ubiquitin ligases (*Venturini et al., 1999*; *Hatakeyama, 2011*; *Herquel et al., 2011*). TRIM33 and its homologs TRIM24 and TRIM28 belong to a subfamily of ubiquitously expressed TRIM proteins that contain a tandem plant homeodomain (PHD) and bromodomain module and function as

**eLife digest** The DNA inside every cell in a human body is the same, and yet the activities that occur within different types of cells can vary greatly. White blood cells, for example, are different from skin cells or liver cells because different genes are active in each type of cell. Molecules called transcription factors and transcriptional cofactors associate with specific DNA sequences to control the activity of nearby genes. It is common for a single transcription factor or cofactor to bind to thousands of sites across the DNA of any cell.

In humans, our immune systems protect us against infectious diseases and from malfunctioning cells that could become cancerous. White blood cells called B cells provide part of this immune defense. These cells help to identify invading bacteria and viruses, and can also develop into memory cells that help the immune system to rapidly recognize, respond to and eliminate a disease if it is re-encountered.

Immature B cells—also known as B lymphoblasts—mature within bone marrow. If any problem occurs in a cell as it matures, that cell is usually programmed to self-destruct in a process called apoptosis. If these cells are not destroyed, they can accumulate in the bone marrow and prevent the production of other immune cells. This leads to a type of cancer called acute lymphoblastic leukemia.

Wang et al. now reveal that TRIM33—a protein that B-lymphoid leukemia cells need to survive—is a transcriptional cofactor that prevents apoptosis. Furthermore, unlike other known transcription factors and cofactors in mammals, TRIM33 binds to an exceedingly small number of sites across the DNA of B cells. In fact, the cancer cell's dependency on the protein is due to TRIM33 associating with just a single binding site.

The role of TRIM33 in B cell leukemia also has potential therapeutic implications. Although it is found in cells throughout the body, Wang et al. found that inhibiting TRIM33 in mice resulted in lower numbers of B cells being produced, but did not affect other tissues. Developing drugs that prevent TRIM33 from working could therefore provide new options for treating leukemia.

transcriptional cofactors (*Venturini et al., 1999*). One role of TRIM33 is in the TGFβ signaling pathway, which occurs through regulation of SMAD4 mono-ubiquitination and/or by functioning as a cofactor for the TFs SMAD2 and SMAD3 (*Dupont et al., 2005*, *2009*; *He et al., 2006*; *Agricola et al., 2011*). TRIM33 is also known to interact with the TF SCL/TAL1 to promote transcription elongation of erythroid-specific genes, which may occur through the recruitment of FACT and P-TEFb complexes (*Bai et al., 2010*, *2013*). However, another study suggests that TRIM33 can form repressive complexes with SCL to block transcriptional activation (*Kusy et al., 2011*). The PHD-bromodomain module of TRIM33 interacts with covalently modified histone tails, which can further stabilize TRIM33 chromatin occupancy and stimulate its E3 ligase activity (*Tsai et al., 2010*; *Agricola et al., 2011*; *Xi et al., 2011*). While several studies have linked TRIM33 to transcriptional control, a genomewide assessment of TRIM33 occupancy has yet to be performed to define the scope of its regulatory function.

## Results

### RNAi screen identifies TRIM33 as a lineage dependency in cancers of B cell origin

In an ongoing effort to identify roles for chromatin regulators as cancer dependencies, we carried out a negative selection RNAi screen in a cell line derived from a mouse model of high risk B cell acute lymphoblastic leukemia (B-ALL), initiated by introducing the BCR-ABL oncogene into *Arf*-null bone marrow cells (*Williams et al., 2006*). 1,126 shRNAs targeting 259 chromatin regulators were assessed individually for effects on the viability of B-ALL cells in culture, which led to the identification of 16 dependencies after applying stringent scoring criteria (*Figure 1A* and *Figure 1—figure supplement 1*). The majority of these factors had been identified in a prior screen that evaluated the requirement of chromatin regulators in MLL-AF9/Nras$^{G12D}$ acute myeloid leukemia (AML) (*Zuber et al., 2011b*), however TRIM33 was identified here as a unique requirement in B-ALL. The six TRIM33 shRNAs used in the screen exhibited a close correlation between knockdown efficiency and loss of B-ALL viability,

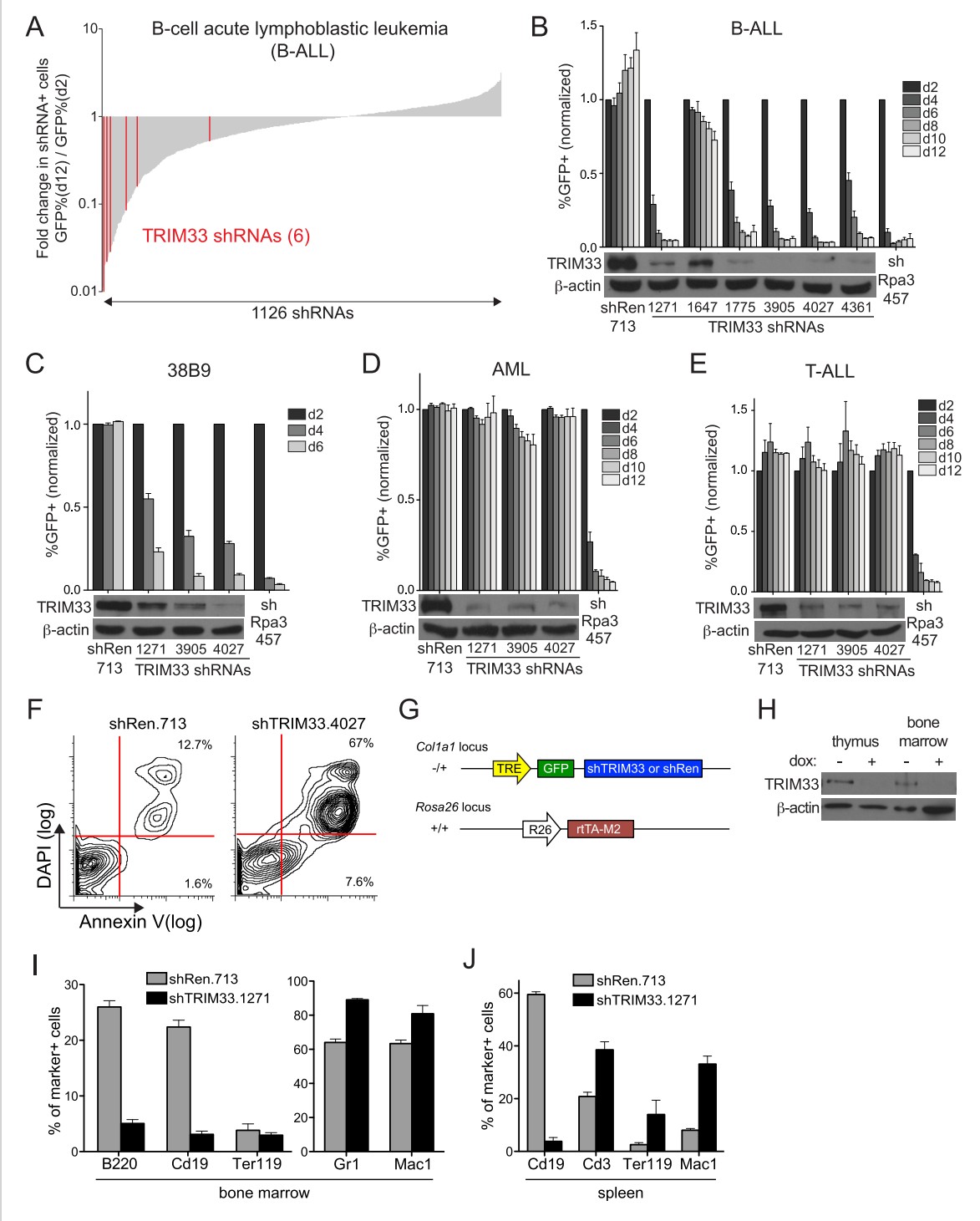

**Figure 1**. RNAi screen identifies TRIM33 as a lineage dependency in cancers of B cell origin. (**A**) Negative selection shRNA screen targeting chromatin regulators in murine B cell acute lymphoblastic leukemia (B-ALL). shRNAs are rank ordered by the fold-change in GFP positivity over 10 days in culture, which represents a competition-based assay in which loss of GFP positivity reflects shRNA-postive cells becoming outcompeted by shRNA-negative cells. (**B–E**) Competition-based assays and Western blotting to evaluate the effect of TRIM33 shRNAs on B-ALL, 38B9, acute myeloid leukemia (AML), or T-cell acute lymphoblastic leukemia (T-ALL) cells. GFP percentages are normalized to day 2 measurements. Results are the average of three biological replicates. (**F**) Annexin V/DAPI staining following transduction of B-ALL cells with the indicated MLS shRNAs on day 3 post-transduction. Representative experiment of three biological replicates is shown. (**G**) shRNA transgenic mouse strategy. TRE: tet(doxycycline) response element; rtTA-M2: reverse tet transactivator M2 variant (tet-on). (**H**) Western blotting performed of indicated tissue lysates prepared from mice treated with dox for 4 weeks.

*Figure 1. continued on next page*

Figure 1. Continued

Representative experiment of three biological replicates is shown. (I–J) Flow cytometry analysis using the indicated antibody stainings of whole bone marrow or spleen. B220 and Cd19: B lymphoid, Ter119: erythroid, Gr1 and Mac1: myeloid, Cd3: T lymphoid. Gating was performed on GFP$^+$/shRNA$^+$ cells prior to quantifying marker positivity. The GFP$^+$ percentage in bone marrow was ~75% and in the spleen was ~15%. Mice were administered dox for 1 week or 4 weeks, with both timepoints giving similar results. Results shown are the average 4 or 5 mice. All error bars in this figure represent S.E.M.

The following figure supplements are available for figure 1:

Figure supplement 1. Hits identified in the shRNA screen and validation experiments in human cell lines.

Figure supplement 2. Additional analysis of TRIM33 shRNA transgenic mice.

suggesting on-target effects (*Figure 1A,B*). Furthermore, a v-Abl transformed B cell progenitor line, 38B9 (*Alt et al., 1984*), and several human B cell cancer lines were also sensitive to TRIM33 knockdown (*Figure 1C* and *Figure 1—figure supplement 1*). In contrast, TRIM33 knockdown led to negligible effects on the viability of MLL-AF9/Nras$^{G12D}$ AML and Notch-mutant T-cell acute lymphoblastic leukemia (T-ALL) (*Figure 1D,E*). Upon knockdown of TRIM33, B-ALL cells underwent apoptosis, as shown by Annexin V/DAPI staining (*Figure 1F*). Taken together, these observations suggest that TRIM33 is essential for the survival of neoplastic B cells.

To compare the TRIM33 requirement in B-ALL with its role in normal tissues, we derived transgenic mice in which a TRIM33 shRNA was expressed from a doxycycline (dox)-regulated promoter (*Figure 1G*) (*Premsrirut et al., 2011*). When crossed with mice that express rtTA-M2 from the ROSA26 promoter, animals exhibited dox-dependent suppression of TRIM33 in multiple tissues (*Figure 1G,H* and *Figure 1—figure supplement 2*). Flow cytometry analysis of the bone marrow and spleen following 1 or 4 weeks of dox administration revealed a selective loss of CD19$^+$ and B220$^+$ B lymphoid cells in TRIM33-deficient animals, while erythroid, myeloid, and T lymphoid populations remained intact or even increased in relative abundance (*Figures 1I,J*). Taking advantage of the reversibility of the dox-regulated shRNA system, we found that B cell production recovered fully in TRIM33-deficient mice upon restoring TRIM33 expression (*Figure 1—figure supplement 2*). Notably, TRIM33 knockdown led to negligible effects on the histological appearance of colon, liver, thymus, and pancreatic tissues and led to no significant changes in animal weight (*Figure 1—figure supplement 2*). Collectively, these findings suggest that B lymphoid cells are uniquely sensitive to TRIM33 inhibition, thus implicating TRIM33 as a lineage dependency in cancers of B cell origin. These results are in agreement with the prior characterization of *Trim33*$^{-/-}$ mice, which were found to exhibit a B lymphocyte deficiency and an expansion of myeloid cells (*Aucagne et al., 2011*; *Kusy et al., 2011*; *Bai et al., 2013*).

## TRIM33 preferentially associates with two lineage-specific enhancers in B lymphoblastic leukemia cells

We next evaluated the mechanism underlying the essential TRIM33 function in B cell neoplasms. To this end, we performed RNA-seq analysis in B-ALL cells following 3 or 4 days of TRIM33 knockdown. This analysis revealed a distribution of gene expression changes, however, we noted that *Bim* and *Atp1b3* were the two most upregulated genes upon TRIM33 depletion (*Figure 2A*). To evaluate whether any of these mRNA changes were due to direct regulation, we performed ChIP-seq analysis in B-ALL cells to evaluate the genomic localization of TRIM33 and various histone modifications that annotate active promoter and enhancer regions. Remarkably, the two strongest sites of TRIM33 enrichment in B-ALL were located 117 kb upstream of *Bim* (in an intron of a non-expressed gene *Acoxl*) and at a site 35 kb upstream of *Atp1b* (*Figure 2B–D*). The other gene expression changes incurred upon TRIM33 knockdown did not correlate with its genomic occupancy (data not shown), suggesting they might be an indirect effect of B-ALL cells initiating an apoptotic response. The TRIM33-occupied regions upstream of *Bim* and *Atp1b3* were enriched for H3K27 acetylation but not for H3K4 tri-methylation, suggesting that these elements are active enhancers (*Rada-Iglesias et al., 2011*) (*Figure 2C,D*). We also observed TRIM33 occupancy at these same two regions in 38B9, AML, and in whole spleen, but not in T-ALL (*Figure 2—figure supplements 1, 2*). A striking attribute of the genomewide pattern of TRIM33 occupancy was its strong bias for a small number of locations, with

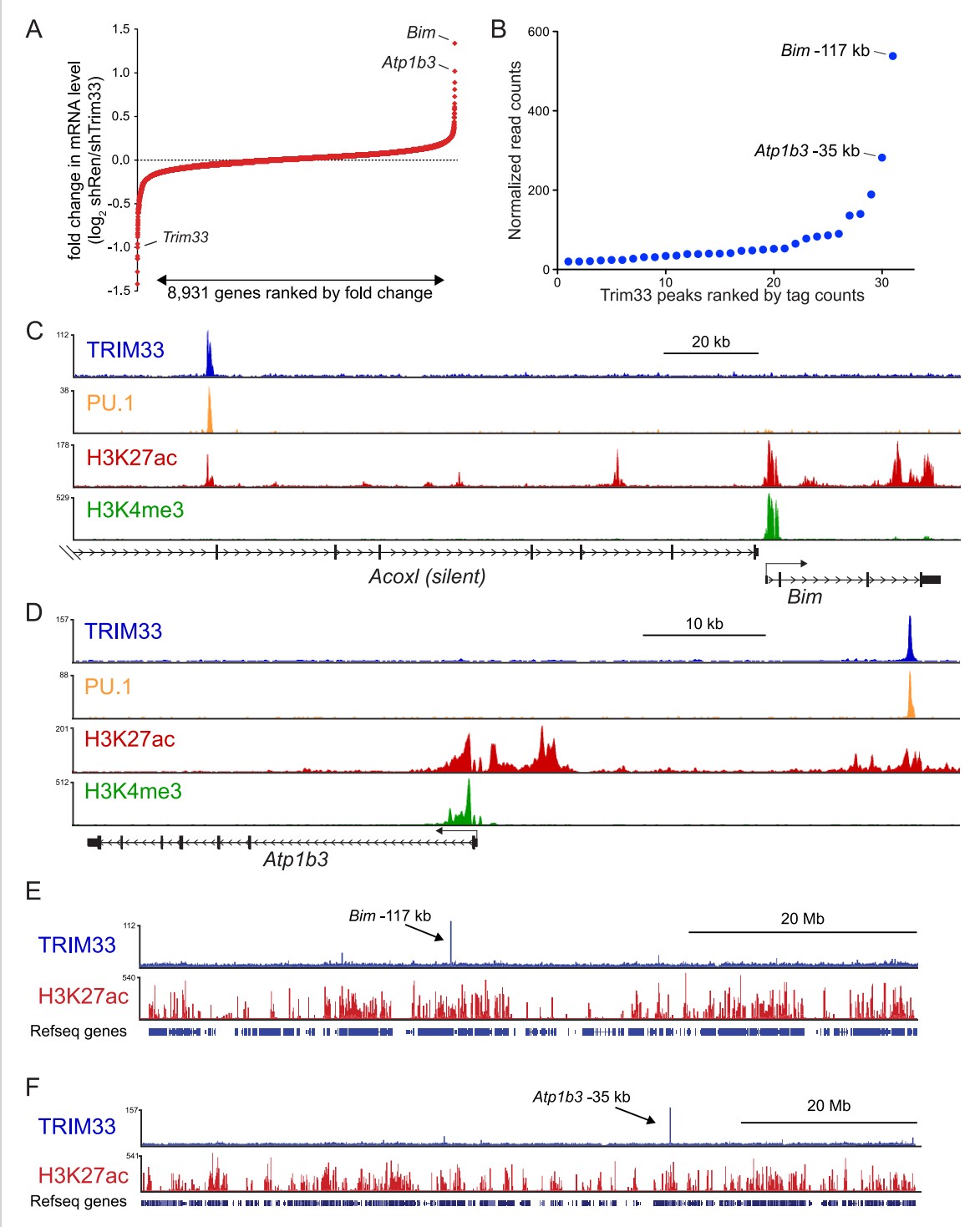

**Figure 2**. TRIM33 preferentially associates with two lineage-specific enhancers in B lymphoblastic leukemia cells. (**A**) RNA-seq analysis of B-ALL cells transduced with shTRIM33.1271. shRNA+/GFP+ cells were sorted on day 3 or 4 post-infection. Plotted is the average fold-change in mRNA level of two independent biological replicates. (**B**) Ranking of TRIM33 occupied sites based on average tag counts obtained from B-ALL ChIP-seq analysis. The 31 regions shown represent the significant reproducible peaks identified in two independent biological replicates. (**C–F**) B-ALL ChIP-seq occupancy profiles using the indicated antibodies. The y-axis reflects the number of cumulative tag counts in the vicinity of each region. Validated transcript models from the mm9 genome assembly are depicted below.

The following figure supplements are available for figure 2:

*Figure 2. continued on next page*

*Figure 2. Continued*

**Figure supplement 1**. (**A–B**) TRIM33 ChIP-seq occupancy profiles at the Bim locus (**A**) and the Atp1b3 locus (**B**) in the indicated cell types.

**Figure supplement 2**. Trim33 ChIP-qPCR analysis in various cell lines.

**Figure supplement 3**. (**A–D**) Comparison of two independent TRIM33 ChIP-seq biological replicates in B-ALL.

**Figure supplement 4**. TRIM33 ChIP-seq analysis in 38B9, AML, and T-ALL.

lower levels of enrichment at other sites across the genome (*Figure 2E,F*, and *Figure 2—figure supplements 3, 4*). This analysis suggests that TRIM33 is concentrated at a small number of sites in the B-ALL genome, with two of these regions correlating with a repressive effect on the expression of nearby *Atp1b3* and *Bim* genes.

## TRIM33 preferentially associates with enhancers harboring an exceptional density of the PU.1 TF

We next pursued the mechanism underlying the profound accumulation of TRIM33 at the *Bim* –117 and *Atp1b3* –35 regions in B-ALL. Using the FIMO motif analysis tool (*Grant et al., 2011*), we found that both of these regions possessed a high density of sequence motifs recognized by PU.1 (17 instances at *Bim* –117 and 14 instances at *Atp1b3* –35), which is an essential hematopoietic TF expressed in B lymphoid and myeloid lineages (*Scott et al., 1994*). In contrast, we observed a much lower density of motifs for other TFs involved in B cell specific transcriptional regulation (E2A, EBF1, or PAX5) (*Figure 3A,B*). Since TRIM33 has been shown previously to interact with PU.1 (*Kusy et al., 2011*), we investigated whether PU.1 promotes TRIM33 recruitment to these regions. ChIP-seq analysis of PU.1 in B-ALL confirmed its association with *Bim* –117 and *Atp1b3* –35, as well as with more than 2600 other genomic sites (*Figure 2C,D* and *Figure 3—figure supplement 1*). Remarkably, *Bim* –117 and *Atp1b3* –35 regions are outliers when compared to other PU.1-occupied sites with regard to the overall density of PU.1 recognition motifs and the overall level of PU.1 enrichment (*Figure 3C* and *Figure 3—figure supplement 1*). The level of the active enhancer chromatin mark, H3K27ac, is not skewed in a similar manner towards these two locations, indicating that these regions are not super-enhancers that exhibit high levels of all regulators (*Figure 3—figure supplement 1*) (*Hnisz et al., 2013*). Using ChIP-qPCR, we also confirmed PU.1 occupancy at these two regions in 38B9 and AML cells, but not in T-ALL (*Figure 3—figure supplement 2*).

These findings raised the possibility that an exceptional density of PU.1 at these two sites might serve as the trigger for TRIM33 recruitment. To evaluate this, we used shRNAs to derive PU.1-deficient B-ALL cells, which displayed normal proliferation and viability (*Figure 3—figure supplement 3*, data not shown). We found that PU.1 knockdown led to a significant loss of TRIM33 occupancy at *Bim* and *Atp1b3* in proportion to the loss of PU.1 (*Figure 3D,E*, and *Figure 3—figure supplement 3*). PU.1-deficient B-ALL cells also harbored reduced H3K27ac at *Bim* –117 and reduced *Bim* mRNA levels, suggesting that PU.1 is responsible for maintaining this enhancer in an active state while, paradoxically, also facilitating TRIM33 recruitment (*Figure 3F* and *Figure 3—figure supplement 3*). Collectively, these results suggest that a high density of PU.1 underlies the skewed distribution of TRIM33 genomic occupancy observed in B-ALL.

## TRIM33 antagonizes PU.1 function to promote B-ALL cell survival

Since PU.1 and TRIM33 appear to have opposing effects on *Bim* expression (*Figure 2A* and *Figure 3—figure supplement 3*), we next evaluated how TRIM33 regulates the function of PU.1 at their co-occupied sites. Interestingly, knockdown of TRIM33 led to an increase in PU.1 occupancy and H3K27 acetylation at *Bim* –117, suggesting that TRIM33 antagonizes PU.1 binding to DNA and suppresses enhancer activity, consistent with prior findings (*Kusy et al., 2011*) (*Figure 3G–I*). This result is significant, since *Bim* –117 is already heavily enriched for PU.1 in B-ALL cells, but PU.1 occupancy becomes even greater when TRIM33 is suppressed. Consistent with the functional importance of PU.1 antagonism by TRIM33, we found that knockdown of PU.1 resulted in a reduced

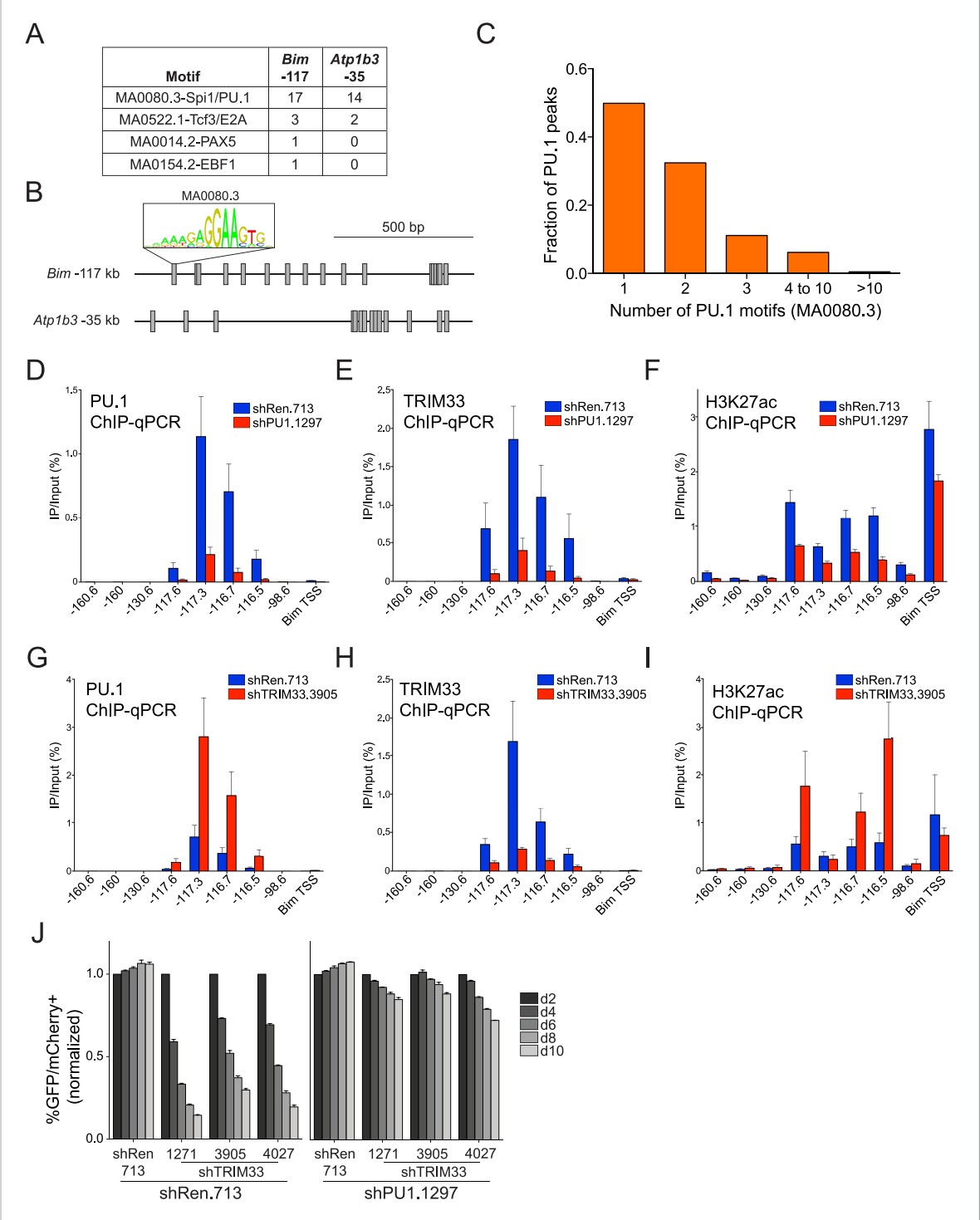

**Figure 3**. TRIM33 is recruited by PU.1 to select enhancers and antagonizes PU.1 function to promote B-ALL survival. (**A**) Motif analysis using the FIMO/MEME algorithm. The table indicates the number of motif matches in each sequence interval. (**B**) Schematic diagram of PU.1 motif (MA0080.3) locations across the indicated *cis* elements. (**C**) Analysis of PU.1 motifs (MA0080.3) counts at all of the 2682 PU.1 peaks identified by ChIP-seq. (**D–I**) ChIP-qPCR analysis at the indicated locations of the *Bim* locus in B-ALL. Labels refer to the kilobase distance relative to the *Bim* TSS. For TRIM33 knockdown experiments (**G–I**), both shRen and shTRIM33 were introduced into B-ALL cells harboring a Bim shRNA. Plotted is the average of three biological replicates. (**J**) Competition-based assay evaluating effect of a PU.1 shRNA on the sensitivity of B-ALL cells to TRIM33 knockdown. shRen or shPU1 linked to GFP were introduced first into B-ALL cells and then were subsequently transduced with shTRIM33 or shRen linked to mCherry. The GFP/mCherry double

*Figure 3. continued on next page*

*Figure 3. Continued*

positive population was measured over time. Results are normalized to day 2 measurements. Plotted is the average of three biological replicates. All error bars in this figure represent S.E.M.

The following figure supplements are available for figure 3:

**Figure supplement 1**. (**A–B**) Ranked ordering of PU.1 and H3K27 acetylation peaks identified using ChIP-seq analysis in B-ALL.

**Figure supplement 2**. (**A–B**) PU.1 ChIP-qPCR analysis at the indicated locations of *Bim* (**A**) and *Atp1b3* (**B**) loci in 38B9, AML, and T-ALL.

**Figure supplement 3**. Additional PU.1 knockdown experiments.

**Figure supplement 4**. Competition-based assay evaluating effect of a PU.1 shRNA on the sensitivity of 38B9 cells to TRIM33 knockdown.

sensitivity of B-ALL and 38B9 cells to the lethal effects of TRIM33 knockdown (*Figure 3J* and *Figure 3—figure supplement 4*). Collectively, these observations suggest that TRIM33 antagonizes PU.1-dependent enhancer activation as part of its essential function in B lymphoblastic leukemia.

## TRIM33 occupancy at a single Bim enhancer accounts for its essential function in B-ALL

*Bim* encodes a BH3-only domain protein with a well-established pro-apoptotic function (*Willis and Adams, 2005*). Therefore, we considered whether repression of *Bim* is the essential role of TRIM33 in B-ALL. For this purpose we suppressed *Bim* using shRNAs and evaluated how this influenced the sensitivity of cells to TRIM33 knockdown (*Figure 4A,B*). Notably, *Bim*-deficient B-ALL cells were completely resistant to the apoptotic effects of TRIM33 shRNAs, indicating that *Bim* is the critical downstream repression target of TRIM33 to maintain cell viability (*Figure 4B*). In contrast, *Atp1b3* knockdown did not alter the sensitivity of B-ALL to TRIM33 suppression (data not shown). To further evaluate whether the essential function of TRIM33 is due to its occupancy at the −117 region, we employed CRISPR-Cas9 to delete this element (*Cong et al., 2013*; *Mali et al., 2013*) (*Figure 4C,D*). Similar to the effects observed upon *Bim* knockdown, three independent B-ALL clones harboring a homozygous deletion of the −117 element did not require TRIM33 for viability (*Figure 4E*). *Bim* knockdown and the deletion of the *Bim* −117 region also rendered 38B9 cells resistant to the lethal effects of TRIM33 suppression (*Figure 4—figure supplement 1*). This indicates that a single genomic binding site at the *Bim* locus accounts for the entire TRIM33 requirement in preventing B-ALL apoptosis.

While both B lymphoid and myeloid leukemia cells harbor PU.1 and TRIM33 at the *Bim* −117 region, it is important to note that knockdown of TRIM33 in AML does not result in apoptosis or *Bim* upregulation (*Figure 1D* and *Figure 4—figure supplement 2*). ChIP-seq analysis of H3K27ac revealed that AML cells possess a large cluster of active enhancers located 66 kb upstream of *Bim*, which is not observed in B-ALL (*Figure 4—figure supplement 2*). Importantly, the level of H3K27ac at this AML-specific enhancer cluster is much greater than that observed at the −117 region. This leads us to speculate that the AML-specific enhancers at the −66 kb region might be the dominant regulatory elements at the *Bim* locus in this cell type. The overall strength of these enhancers in AML may diminish the functional significance of the −117 element, thereby rendering TRIM33 dispensable for AML survival.

## Discussion

Here we provide evidence that TRIM33 prevents apoptosis in murine B-ALL cells by blocking enhancer-mediated *Bim* activation. The role of TRIM33 as a negative regulator of enhancer activity is reminiscent of LSD1, a histone demethylase that has been shown previously to suppress active enhancers during embryonic stem (ES) cell differentiation (*Whyte et al., 2012*). However, a key distinction we make here is that LSD1 suppresses thousands of enhancers in ES cells, while TRIM33 preferentially accumulates at a small number of regulatory elements in the B-ALL genome. We trace this unusual enhancer selectivity of TRIM33 in B-ALL to the density of PU.1, a TF which has been shown previously to associate with TRIM33 (*Kusy et al., 2011*). This binding interaction is likely to be cooperative on motif-rich DNA elements, which would account for the biased accumulation of TRIM33

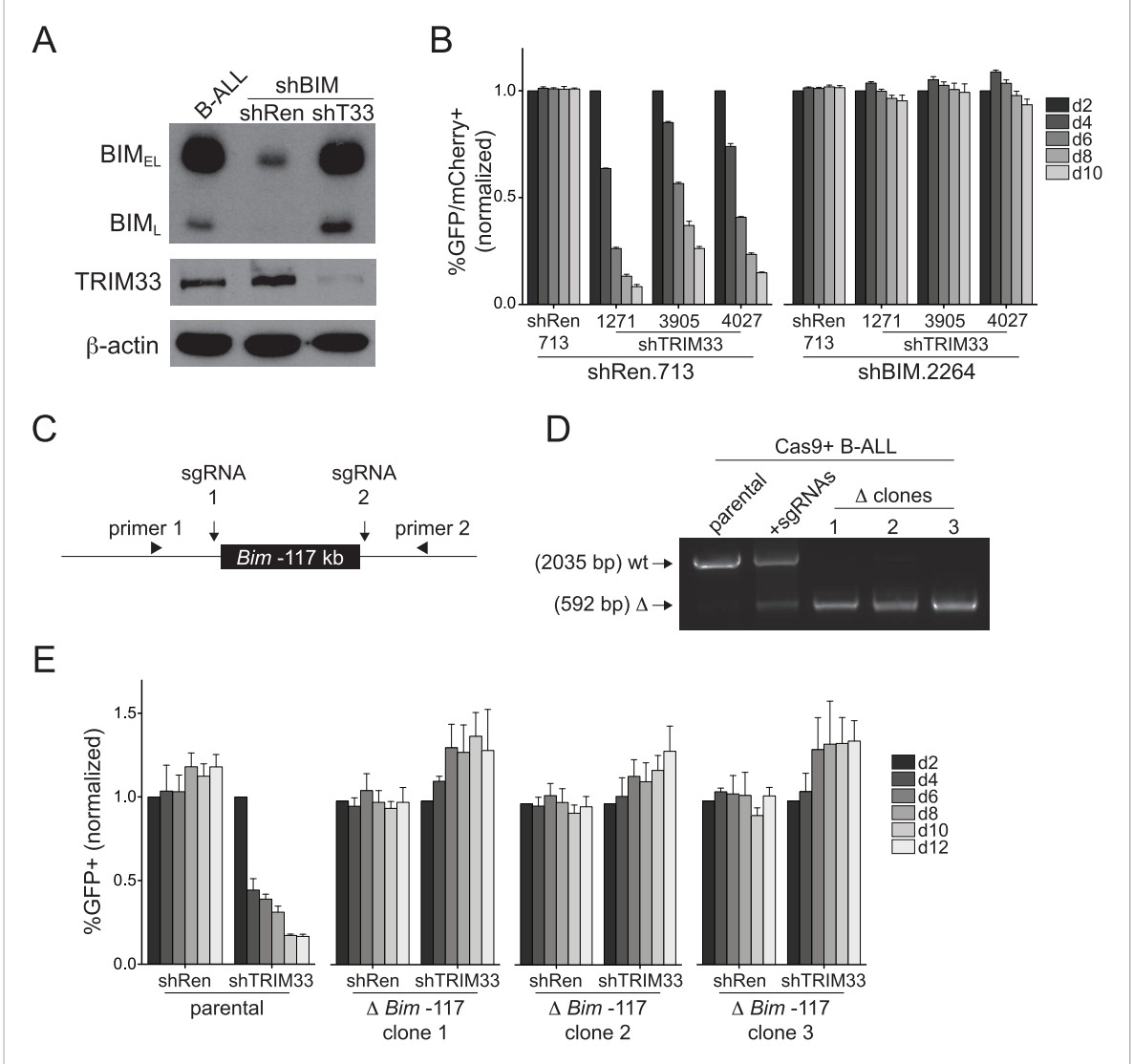

**Figure 4**. TRIM33 occupancy at a single Bim enhancer accounts for its essential function in B-ALL. (**A**) Western blot performed on extracts prepared from B-ALL following transduction with the indicated shRNAs. Labeled are two of the known Bim isoforms. TRIM33.3905 shRNA was used. (**B**) Competition-based assay evaluating the effect of Bim knockdown on the sensitivity of B-ALL cells to TRIM33 knockdown. shRen or shBim linked to GFP were introduced first into B-ALL cells and then were subsequently transduced with shTRIM33 or shRen linked to mCherry. The GFP/mCherry double positive population was measured over time. Results are normalized to day 2 measurements. Plotted is the average of three biological replicates. (**C**) Experimental design for generating a homozygous deletion of the *Bim* −117 region using CRISPR-Cas9. sgRNAs were designed to cut at locations flanking the TRIM33 binding site to delete the intervening 1.4 kb. Location of PCR primers used for genotyping are indicated. (**D**) Genotyping PCR to track the CRISPR-based deletion. Parental refers to Cas9+ B-ALL cells. +sgRNAs refers to Cas9+ B-ALL cells co-transduced with two sgRNAs targeting the *Bim* −117 element linked to mCherry reporters. Three clonal lines were derived by limiting dilution. (**E**) Competition-based assay evaluating the effect of the *Bim* −117 deletion on the sensitivity to TRIM33 knockdown. B-ALL cells were transduced with indicated shRNAs linked to GFP reporters. TRIM33.3905 shRNA was used. Results are normalized to day 2 measurements. Plotted is the average of three biological replicates. All error bars in this figure represent S.E.M.

The following figure supplements are available for figure 4:

**Figure supplement 1**. Additional experiments evaluating the role of Bim in 38B9 cells.

**Figure supplement 2**. Additional experiments evaluating the role of TRIM33 in AML and T-ALL.

at enhancers harboring an exceptional PU.1 density. Our ChIP-seq analysis of TRIM33 in myeloid, B lymphoid, and T lymphoid leukemias suggests that other TFs are also likely to recruit TRIM33, such as TCF3/E2A (*Figure 2—figure supplement 4*). To our knowledge, TRIM33 is the first example of

a mammalian transcriptional regulator that performs an essential function through a single genomic binding site.

The selective occupancy of TRIM33 within the active enhancer landscape may account for the minimal phenotypes observed in TRIM33-deficient mice. This is in contrast to the effects of suppressing BRD4, a more general cofactor that is essential for the maintenance of many cancer types and for the homeostasis of several normal tissues (*Bolden et al., 2014*; *Shi and Vakoc, 2014*). Remarkably, it is the localization of TRIM33 to one lineage-specific enhancer upstream of *Bim* that renders B-ALL cells hypersensitive to loss of TRIM33. The similar sensitivity of normal and transformed B cells to TRIM33 inhibition implies that the regulatory mechanisms defined here in B cell leukemia are likely to have evolved to control normal B cell development. Interestingly, *Bim* is upregulated as a means to eliminate auto-reactive B cells and to prevent autoimmune disease, suggesting a rationale for precise modulation of enhancer activity at the *Bim* locus by TRIM33 to regulate adaptive immunity (*Bouillet et al., 1999*; *Enders et al., 2003*).

In humans, a reversible depletion of normal B cells is a well-tolerated side effect of the anti-CD20 antibody rituximab, which is an efficacious therapy in mature B cell neoplasms and in autoimmune disease (*Coiffier et al., 2002*; *Edwards et al., 2004*). Hence, our findings also implicate TRIM33 as a promising therapeutic target, since one of the major phenotypic abnormalities in mice experiencing systemic TRIM33 inhibition is a selective and reversible B lymphocyte deficiency. Targeting of TRIM33 is particularly attractive since other bromodomain modules (e.g., BRD4) have proven to be amenable to direct chemical inhibition (*Prinjha et al., 2012*). However, we observe a modest expansion of myeloid cells upon short durations of TRIM33 suppression in vivo and *Trim33*$^{-/-}$ mice have been shown to develop chronic myelomonocytic leukemia (*Aucagne et al., 2011*). Thus, long-term inhibition of TRIM33 might have pro-tumorigenic effects in certain tissues. Nonetheless, our findings justify consideration of TRIM33-inhibition as a therapeutic approach in B cell neoplasms, which would be expected to unleash an apoptosis-promoting enhancer element.

## Materials and methods

### Plasmids

For competition-based assays in murine cells, the LMN-GFP or LMN-mCherry shRNA retroviral vector were used (MSCV-miR30-shRNA-PGKp-NeoR-IRES-GFP/mCherry). When drug selection was required, some experiments were performed using MLP-GFP (MSCV-miR30-shRNA-PGKp-PuroR-IRES-GFP). For Annexin V experiments, the MLS vector was used (MSCV-miR30-shRNA-SV40p-GFP) to obtain a higher infection efficiency. For competition assays in human cell lines, the MLS-E vector was used (MSCV-miRE-shRNA-SV40p-GFP) (*Fellmann et al., 2013*). All of the cloning procedures were performed using the In-Fusion cloning system (#638909; Clontech, Mountain View, CA).

### Cell culture and virus production

Murine B-ALL (driven by BCR-ABL and p19$^{Arf}$ inactivation) (*Williams et al., 2006*) and 38B9 (*Alt et al., 1984*) were cultured in RPMI supplemented with 10% fetal bovine serum (FBS), 1% penicillin/streptomycin, and 0.055 mM 2-Mercaptoethanol. Derivation of MLL-AF9/Nras$^{G12D}$ AML cells is described in *Zuber et al. (2011a)*. The T-ALL cell line was provided by I Aifantis (NYU) and was derived from TAL transgenic/HEB heterozygous mice with an acquired truncation of Notch1. Murine AML and T-ALL cells and human REH, SU-DHL, MM1S, and JURKAT cell lines were cultured in RPMI/10% FBS/1% penicillin/streptomycin.

The retroviral and lentiviral packaging cells Ecotropic Plat-E cells (*Morita et al., 2000*) and HEK293T cells were cultured in DMEM supplemented with 10% FBS and 1% penicillin/streptomycin. All retroviral packaging was performed according to established protocols (*Morita et al., 2000*). Virus was collected 48–72 hr post-transfection.

### Negative selection RNAi screen

A previously described LMN shRNA library targeting chromatin regulatory proteins was used (*Zuber et al., 2011b*). Additional shRNAs were included in the library to allow comprehensive targeting of all bromodomain-containing proteins. shRNA plasmids from this library were arrayed individually in 96 well plates at 50 ng/μl for retrovirus preparation using Ecotropic Plat-E cells as described (*Morita et al., 2000*). Viral transduction of B-ALL cells was also performed in 96 well plates, followed by

flow-cytometry-based measurement of GFP positivity using a Guava Easycyte instrument (Millipore, Billerica, MA). Measurements were performed on day 2 and day 12 post-transduction. Renilla and Rpa3 control shRNAs were included with each 96 well plate as negative and positive controls, respectively. If an shRNA was introduced into B-ALL cells with a day 2 GFP percentage below 5%, this sample was discarded from the screen. The lower limit on GFP% measurements on day 12 was arbitrarily set at 0.2%. A single instance of Renilla and Rpa3 control shRNAs are included in the screen histogram shown in *Figure 1A*. The screen data can be found in *Supplementary file 1*.

## Competition assay to measure cellular effects of shRNA

To evaluate the effect of shRNAs on murine cell proliferation (e.g., in *Figure 1A–E*), cultures were retrovirally transduced with the indicated LMN-GFP shRNA vectors, followed by measurement of the GFP percentages over a time course post-infection using a Guava Easycyte flow cytometer. For double shRNA competition-based assays (e.g., *Figures 3J, 4B*), cells were first transduced with MLP-GFP (Ren, Bim, or PU.1 shRNA) and then were selected with puromycin (1 µg/ml). Cells were subsequently transduced with LMN-mCherry (Ren or TRIM33 shRNAs) and double GFP/mCherry positivity was tracked using a BD LSR2 flow cytometer. For human competition assays in *Figure 1—figure supplement 1*, the MLS-E vector was used and GFP was tracked using a Guava Easycyte instrument. shRNA sequences can be found in *Supplementary file 2*.

## Annexin V staining for cell death analysis

In order to quantify the cell death effect induced by Renilla or TRIM33 shRNAs, B-ALL cells were transduced at a high efficiency (>95% GFP) using the MLS-GFP vector. After 3 days post-infection, cells were spun down, washed with PBS and resuspended in 1× binding buffer and incubated with APC Annexin V antibody (as described by manufacturer protocol, #550475; BD Pharmingen, San Diego, CA) and DAPI, followed by BD LSRII flow cytometry analysis. GFP gating was not performed. The three independent TRIM33 shRNAs revealed similar results. The data analysis was performed using Flowjo software.

## Transgenic animal studies

All experimental mouse procedures were approved by the Cold Spring Harbor Animal Care and Use Committee. Renilla and TRIM33 shRNA targeted ES cells were derived by Mirimus (Cold Spring Harbor, NY) as described (*Premsrirut et al., 2011*; *Dow et al., 2012*), using site-specific integration of the shRNA cassette into the *Col1a1* locus of KH2 ES cells (*Beard et al., 2006*). 8–10 week old mice heterozygous for alleles of shRen.713 or shTRIM33 and homozygous for ROSA26-rtTA-M2 alleles were treated with dox drinking water (2 mg/ml doxycycline and 2% sucrose) and dox food (625 mg/kg) for 7 or 28 days. Fluorescent markers (eGFP for shRenilla and TurboGFP for shTRIM33) were indicators of shRNA-expressing cells following doxycycline treatment. The animal health and body weight were monitored during the time course of experiments. Primers used for genotyping can be found in *Supplementary file 2*.

To measure the impact of TRIM33 knockdown on hematopoietic lineages, bone marrow and spleen tissues were harvested from 8–10 week old transgenic mice following the administration of dox for 7 or 28 days. A single cell suspension of harvested samples was treated with ACK buffer (150 mM $NH_4Cl$, 10 mM $KHCO_3$, 0.1 mM EDTA) to lyse the red blood cells. The sample was then stained with hematopoietic lineage marker antibodies (1:200) in FACS buffer (1× PBS with 5% FBS and 0.05% $NaN_3$) for 30 min and then subjected to BD LSRII flow cytometry analysis. Gating was performed on live cells (FSC/SSC) and on GFP+ cells prior to quantifying marker positivity. The GFP percentage in total bone marrow was approximately 75% on dox for shRen and shTRIM33 mice. The GFP percentage in the total spleen was approximately 15% for shTRIM33 and shRen mice on dox. This heterogeneity in GFP expression in these different tissues requires GFP gating to accurately define hematopoietic phenotypes using flow cytometry. The data analysis was performed using Flowjo software.

To evaluate the reversibility of the TRIM33 knockdown phenotype in bone marrow B lymphoid cell, animals were placed on dox for 7 days and a second cohort was placed on dox for 7 days followed by removal of dox for 21 additional days. B220 and Cd19 marker staining was performed as described above, but the GFP marker was not gated upon since GFP expression is extinguished upon removing dox. This accounts for the weaker B cell phenotype in this 7 day dox treatment from the findings presented in *Figure 1I*, in which GFP gating was performed.

For immunohistochemistry analysis of TRIM33 expression, transgenic mice were maintained on dox for 28 days. Mouse tissues were fixed overnight in 10% neutral buffered formalin and then washed with PBS. Samples were then processed using Shandon Excelsior Tissue processor and embedded in paraffin, 6 micron sections were then mounted onto VWR Superfrost Plus slides.

## Western blotting

For western blotting, whole cell lysates were prepared by direct lysis using SDS-PAGE sample buffer and about 50,000 cell equivalents were loaded into each lane. Samples were then separated by SDS-PAGE electrophoresis and transferred to nitrocellulose for detection using antibodies. For western blotting of mouse tissues, samples were harvested and lysed in RIPA buffer (25 mM Tris-HCl pH 7.6, 150 mM NaCl, 1% NP-40, 1% sodium deoxycholate and 0.1% SDS) using a dounce homogenizer, then briefly sonicated and incubated on ice for 15 min. Pellets were spun down and the supernatant was collected and quantified using a Bradford assay. 20 µg of protein were loaded into each gel lane.

## Immunohistochemistry

Deparaffinization of samples was done with Xylene (2 × 10 min) followed by ethanol rehydration (100%-2 × 5 min, 95% and 75% 2 min each) and washed in distilled water. 3% Hydrogen Peroxide was used for blocking endogenous peroxidase, followed by another wash. Antigen retrieval was done with Citrate buffer in an Electric Pressure Cooker then washed in TBS. Samples were then blocked with 5%NHS, 1%BSA in TBS for 1 hr at room temperature and incubated with TRIM33 antibody (1:500/1:1000) overnight at 4°C. Followed by washing with TBS and incubated with Vector mp-7401 anti-rabbit secondary antibody for 30 min and then washed again. Slides were then incubated with Vector ImmPACT DAB, SK-4105 for 3 min and then rinsed in water. Counterstaining was performed with Hematoxylin and coverslipping was performed with Surgipath mounting medium prior to analysis.

## Chromatin immunoprecipitation

Cells were crosslinked with 1% formaldehyde for 20 min at room temperature and then quenched with 0.125 M glycine. Samples were sonicated and incubated with 5 µg of antibody overnight and then precipitated using Protein A Dynabeads (cat #10002Dl; Life Technologies, Grand Island, NY). For TRIM33/PU.1 ChIP in B-ALL, 50 million cells were used for each ChIP conditions with 5 µg antibody. For histone modification ChIP experiments in B-ALL, 20 million cells were used for each ChIP conditions with 2 µg antibody. For whole spleen ChIP, 6–8 week old mice were sacrificed and a suspension of 30 million splenocytes was prepared for ChIP with 5 µg antibody. ChIP-qPCR was performed on the reverse crosslinked DNA samples as previously described (*Steger et al., 2008*). All of the results were quantified by qPCR using SYBR green (ABI, Grand Island, NY) on an ABI 7900HT. Each IP signal was quantified based on an input standard curve dilution series of the pre-immunoprecipitated genomic DNA (IP/Input) to normalize for the differences of total amount of cells subjected for ChIP and for the different amplification efficiencies of various primer sets.

## RT-qPCR analysis

Total RNA was extracted from cells using Trizol reagent according to the manufacturer's instructions. Upon isolating RNA, DNase I was treated to eliminate contaminating genomic DNA. For cDNA synthesis, Q-Script cDNA SuperMix (Quanta BioScience, Gaithersburg, MD) was used. All results were quantified by qPCR performed using SYBR green (ABI) on an ABI 7900HT using the delta Ct method using *Gapdh* as a control gene.

All ChIP and RT-qPCR primers used in this study can be found in *Supplementary file 2*.

## ChIP-seq and RNA-seq library preparation

ChIP DNA was purified using a QIAquick Gel Extraction Kit (Qiagen, Valencia, CA) and ChIP-Seq libraries were constructed using TruSeq ChIP Sample Prep Kit (Illumina) following the manufacturer's instructions. The quality of each library was determined by Bioanalyzer analysis using the High Sensitivity chip (Agilent). Two independent biological replicates were performed for each ChIP-seq experiment. Barcoded libraries were sequenced in a multiplexed fashion with two to six

libraries at equal molar ratio, with single end reads of 50 bases. For RNA-seq, total RNA was extracted using Trizol reagent (Invitrogen, Carlsbad, CA). Libraries were constructed using the TruSeq sample Prep Kit V2 (Illumina, San Diego, CA) according to the manufacturer's instructions. DNA libraries were sequenced using an Illumina HiSeq 2000 platform.

## RNA-seq analysis

The obtained reads were trimmed into 28 base reads corresponding to 9th to 36th position from the 5′ ends of the reads. These reads were mapped to the mouse genome (mm9) using Tophat software allowing no mismatch, then differentially expressed genes were analyzed by using Cuffdiff software. During this step, structural RNAs (e.g., ribosomal or mitochondrial RNA) were masked. To calculate relative fold change to control, only genes with expression cutoff above 5 reads per million per kilobase (RPKM) and OK test status were considered. Average RPKM from two biological replicates of each control (Ren) and TRIM33 shRNA expressed samples was then used to calculate fold change with log2 scale.

## ChIP-seq analysis

The sequence reads were of 36 or 50 bp in length and mapped to the reference murine genome assembly NCBI37/mm9 using Bowtie. Alignments were performed using the following criteria: -m1, -v2. To identify ChIP-Seq peaks, we used the MACS version 1.4.0 beta (Model based Analysis of ChIP-Seq) peak finding algorithm by using a p value threshold of enrichment of 1e-5 as a cut-off. For AML, 38B9, and T-ALL, we also implemented a 10-fold IP/Input enrichment ratio as a filtering criteria. To identify reproducibly enriched regions of TRIM33 ChIP-Seq from two biological replicates, called peaks from MACS were compared and intersected. If the peaks showed at least 1 bp overlap between replicates, they were considered as reproducible. To calculate average tag counts from reproducible peaks, tag counts were normalized to total mapped reads, and further ranked by tag counts.

All ChIP-seq and RNA-seq data from this study can be found at the GEO accession super-series GSE66234.

## FIMO/MEME motif analysis

Sequence coordinates from TRIM33 peak calling using MACS were used to obtain sequences of Bim-117 and Atp1b3 regions. Analysis was performed at the FIMO/MEME website: http://meme.nbcr.net/meme/tools/fimo using a p-value output threshold of 1E-4 and motif file JASPAR_CORE_2014_VERTEBRATES.MEME from the Jaspar database (*Mathelier et al., 2014*).

For the unbiased discovery of motifs at TRIM33 enriched regions, 400 bp core sequences centered around reproducible TRIM33 peaks were submitted to the browser-based MEME-ChIP analysis (http://meme.nbcr.net/meme/tools/meme-chip).

## CRISPR-Cas9 targeted deletion of *Bim* −117

The MSCV-hCas9-PGK-Puro construct was derived by cloning the N terminal 3× FLAG tag human-codon optimized Cas9 cDNA (#49535; Addgene) into the MSCV-PGK-Puro vector (#634401; Clontech). The U6-sgRNA-EFS-mCherry vector was constructed using the lentiviral backbone from lentiCRISPR (#49535; Addgene). All sgRNAs were designed using http://crispr.mit.edu/ and cloned into the U6-sgRNA-EFS-mCherry vector following published protocols (*Ran et al., 2013*). For sgRNA lentivirus production, HEK293T cells were transfected with sgRNA:pVSVg:psPAX2 plasmids in a 4:2:3 ratio by using PEI reagent (#23966; Polysciences, Warrington, PA) following standard procedures. To generate CRISPR competent line, parental B-ALL and 38B9 cells were transduced with MSCV-hCas9-PGK-Puro construct followed by puromycin selection (1 μg/ml). To generate lines with a homozygous deletion of *Bim* −117, two sgRNAs were designed to target flanking sequences of the −117 element and were retrovirally co-transduced into B-ALL or 38B9 cells, followed by isolation of clonal lines by limiting dilution. Genomic DNA was prepared from each clone using QiAamp DNA mini kit (#51304; Qiagen) following the manufacturer's instructions, followed by PCR-based screening for clones harboring a homozygous deletion, using primers that span the *Bim* enhancer region. The −117 deletion of individual clones was further verified by Sanger sequencing. sgRNA and genotyping primer sequences can be found in *Supplementary file 2*.

## Antibodies

Anti-TRIM33 antibody (A301-060A; Bethyl Laboratories, Montgomery, TX), Anti-PU.1 antibody (#2266; Cell Signaling, Beverly, MA or sc-352; Santa Cruz, Dallas, TX), Anti-Bim antibody (#2819; Cell Signaling), Anti-H3K27ac antibody (ab4729; Abcam, Cambridge, MA), Anti-H3K4me3 antibody (07-473; Millipore), Anti-ß-actin HRP antibody (#A3854; Sigma, Ronkonkoma, NY), APC anti-mouse B220 (#103212; BioLegend, San Diego, CA), APC anti-mouse CD-19, APC anti-mouse Mac-1/Cd11b (#101211; BioLegend), APC anti-mouse Ly-6G/Gr-1 (#17-5931; eBioscience, San Diego, CA), APC anti-mouse TER-119 (#116212; BioLegend), APC anti-mouse CD-3 (#100209; BioLegend).

## Acknowledgements

We thank Jerry Pelletier for providing the murine B-ALL cell line and Michael Atchison for providing 38B9 cells. John Erby Wilkinson for analysis of histological sections. This work was supported by Cold Spring Harbor Laboratory National Cancer Institute Cancer Center Support grant CA455087 for developmental funds and shared resource support. Additional funding was provided by the Alex's Lemonade Stand Foundation and the V Foundation. CRV is supported by a Burroughs-Wellcome Fund Career Award and National Institutes of Health grant NCI RO1 CA174793. SK is supported by JSPS fellowship. JSR is supported by the Martin Sass Foundation. ASB is supported by NCI F30 CA186632. AFH is supported by a Boehringer Ingelheim Fonds PhD Fellowship.

## Additional information

### Funding

| Funder | Grant reference | Author |
|---|---|---|
| National Cancer Center (NCC) | Cold Spring Harbor Laboratory CA455087 | Christopher R Vakoc |
| Alex's Lemonade Stand Foundation for Childhood Cancer | A award | Christopher R Vakoc |
| Burroughs Wellcome Fund (BWF) | Career award for medical scientist | Christopher R Vakoc |
| Japan Society for the Promotion of Science (JSPS) | postdoctoral fellowship | Shinpei Kawaoka |
| SASS Foundation for Medical Research | postdoctoral fellowship | Jae-Seok Roe |
| National Cancer Institute (NCI) | NCI F30 CA186632 | Anand S Bhagwat |
| Boehringer Ingelheim Fonds (BIF) | PhD fellowship | Anja F Hohmann |
| V Foundation for Cancer Research | | Christopher R Vakoc |
| National Institutes of Health (NIH) | NCI RO1 CA174793 | Christopher R Vakoc |

The funders had no role in study design, data collection and interpretation, or the decision to submit the work for publication.

### Author contributions

EW, ASB, CRV, Conception and design, Acquisition of data, Analysis and interpretation of data, Drafting or revising the article; SK, Conception and design, Acquisition of data, Analysis and interpretation of data; J-SR, Analysis and interpretation of data, Drafting or revising the article; JS, JBK, Conception and design, Analysis and interpretation of data, Drafting or revising the article; AFH, YX, Conception and design, Acquisition of data, Drafting or revising the article; YS, Acquisition of data, Analysis and interpretation of data, Drafting or revising the article

### Ethics

Animal experimentation: All experimental mouse procedures were approved by the Cold Spring Harbor Animal Care and Use Committee, protocol #12-09-23. This institution has an Animal Welfare Assurance on file with the Office for protection of research risks. The assurance number is A3280-01.

## Additional files

### Supplementary files

• Supplementary file 1.  B-ALL shRNA screen data.

• Supplementary file 2.  Sequences of primers, shRNAs, and sgRNAs used in this study.

### Major datasets

The following datasets were generated:

| Author(s) | Year | Dataset title | Dataset ID and/or URL | Database, license, and accessibility information |
|---|---|---|---|---|
| Wang E, Kawaoka S, Roe JS, Shi J, Hohmann A, Xu Y, Bhagwat A, Suzuki Y, Kinney J, Vakoc CR | 2015 | The transcriptional cofactor TRIM33 prevents apoptosis in B lymphoblastic leukemia by deactivating a single enhancer | http://www.ncbi.nlm.nih.gov/geo/query/acc.cgi?acc=GSE66234 | Publicly available at NCBI Gene Expression Omnibus (GSE66234). |
| Shi J, Kawaoka S, Zhu Z, Kendall J, Wigler MA, Vakoc CR | 2013 | Role of SWI/SNF in acute leukemia maintenance and enhancer-mediated Myc regulation (RNA-seq) | http://www.ncbi.nlm.nih.gov/geo/query/acc.cgi?acc=GSE52623 | Publicly available at NCBI Gene Expression Omnibus (GSE52623). |

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
