## [Decision Letter]

Thank you for sending your work entitled “Trim33 prevents apoptosis in B lymphoblastic leukemia by decommissioning a single enhancer” for consideration at *eLife*. Your article has been favorably evaluated by Charles Sawyers (Senior editor) and 3 reviewers, one of whom is a member of our Board of Reviewing Editors.

The Reviewing editor and the other reviewers discussed their comments before we reached this decision, and the Reviewing editor has assembled the following comments to help you prepare a revised submission:

The manuscript by Wang et al. describes a lineage-specific epigenetic dependency, namely on Trim33. Many B-lineage cancers and autoimmune diseases continue to be urgent clinical problems, increasing the impact of this manuscript. The authors define a novel mechanism, where highly clustered binding sites for the transcription factor PU.1 at enhancer sites are counteracted by Trim33. This disproportionally affects 2 genes including the apoptosis inducer *Bim*. The authors demonstrate functional importance of *Bim* and functional importance of a single enhancer element upstream of *Bim*. Thus, virtually all of the oncogenic effects of TRIM33 can be attributed to binding to a single site in the genome, a remarkable discovery.

The reviewers agreed that the paper was very interesting and well crafted, but there was some discussion about limited impact due to the fact that most mechanistic work was done in a single cell line, and that the mechanism of action of TRIM33 is largely unexplored. The reviewers agreed that further insight into the mechanism of action was not needed for a Short Report, but there was consensus on requiring validation of key mechanistic experiments in a second cell line.

More specifically, the authors show a unique higher density of PU.1 at the *Bim*-117 enhancer and antagonism of PU.1 by TRIM33 recruitment to this enhancer in B-cell derived leukemic and progenitor cell lines. However, while they show the initial phenotype for several B cell lines, B-ALL, 38B9 (Figure 1) and other B cell cancer lines (Figure 1—figure supplement 1), all mechanistic work (except for TRIM33 occupancy) is done in only one cell line (B-ALL). The authors should reproduce their mechanistic studies in a second cell line (e.g., 38B9). Otherwise the novelty of this work is somewhat limited: it is known that PU.1 binds enhancers, and that TRIM33 interacts with PU.1. The reviewers agreed on the need to delineate the PU.1/TRIM33/*BIM* pathway in 38B9 cells, with the exception of removal of the enhancer at the *BIM* locus, which was thought to be too involved and would delay publication.

Minor points:

1) In the Discussion the authors claim TRIM33 as a therapeutic target “… since the major phenotypic abnormality in mice experiencing systemic Trim33 inhibition is a selective and reversible B lymphocyte deficiency.” The authors do not address the other hematopoietic lineage phenotypes previously described by Aucagne et al., J. Clin. Invest., 2011 and Bai et al., Dev. Biol., 2013, most importantly the CMML phenotype. That is surprising as the authors also see an expansion of the granulocytic compartment after 1 week (BM, Figure 1), and more dramatically 4 weeks (spleen, Figure 1). Please discuss this.

2) If PU.1 is a lineage-specific factor for many myeloid lineages, why would TRIM33 antagonize PU.1 specifically in B (progenitor) cells? What about PU.1 density at the *Bim* enhancers in AML or T-ALL cell lines (the authors only show TRIM33 occupancy in those)? One would expect that either PU.1 density is low or that *Bim* does not play a key role in these cell lines. This should be discussed.

3) How do the authors imagine the interplay of TRIM33 and PU.1 in normal B cell differentiation? Is this mechanism active in vivo? The authors show that in the mouse model depleted for Trim33 B-lymphoid cells are selectively lost. They do not address whether apoptosis in the model is due to PU.1-TRIM33 antagonism. In fact, they show that TRIM33 binds to the *Bim* enhancer in normal spleen. However, work by Oliver et al., J. Exp. Med., 2004 shows that *Bim* is expressed in B-cell precursors. Is this also true in the authors’ system? If so, how do they explain *Bim* expression despite TRIM33 occupancy at the enhancer? What are the absolute *Bim* levels compared to those in the B-ALL model?

4) The fact the GFP-gated cells in vivo show more efficient B-lineage depletion than non-GFP-gated cells (and that there are non-GFP cells present under induced conditions in the marrow) suggests to me that the mouse strain with inducible shRNA/GFP has variegated expression in the B-linage or even whole BM (position effect). I believe this does not affect the manuscript's message, but the authors should clarify.

5) The authors allude to other factors with documented peak number: MyoD ∼39,000 and Pax5 ∼20,000 sites (9; 33). In a side by side comparison, using the same peakfinder and same parameters including p-value cutoff, how many significant peaks are there for Trim33 in the p190-leukemia cells? It might be that not the number of peaks is different from characterized transcriptional regulators but the existence of the unusual outliers including *Bim*. Please clarify.

---

## [Author Response]

*The reviewers agreed that the paper was very interesting and well crafted, but there was some discussion about limited impact due to the fact that most mechanistic work was done in a single cell line, and that the mechanism of action of TRIM33 is largely unexplored. The reviewers agreed that further insight into the mechanism of action was not needed for a Short Report, but there was consensus on requiring validation of key mechanistic experiments in a second cell line*.

*More specifically, the authors show a unique higher density of PU.1 at the Bim-117 enhancer and antagonism of PU.1 by TRIM33 recruitment to this enhancer in B-cell derived leukemic and progenitor cell lines. However, while they show the initial phenotype for several B cell lines, B-ALL, 38B9 (*Figure 1*) and other B cell cancer lines (*Figure 1—figure supplement 1*), all mechanistic work (except for TRIM33 occupancy) is done in only one cell line (B-ALL). The authors should reproduce their mechanistic studies in a second cell line (e.g., 38B9). Otherwise the novelty of this work is somewhat limited: it is known that PU.1 binds enhancers, and that TRIM33 interacts with PU.1. The reviewers agreed on the need to delineate the PU.1/TRIM33/*BIM *pathway in 38B9 cells, with the exception of removal of the enhancer at the* BIM *locus, which was thought to be too involved and would delay publication*.

Our revised manuscript now includes several additional mechanistic experiments evaluating the PU.1/TRIM33/*BIM* pathway in 38B9 cells, which includes ChIP-seq analysis of TRIM33 in 38B9 (Figure 2—figure supplement 1). This epigenomic analysis highlights a similar binding pattern for TRIM33 at the *Bim* locus in this cell type. We find a larger number of peaks for TRIM33 in 38B9, however, *Bim*-117 and Atp1b3-35 are among the most profoundly enriched in 38B9, similar to the observations in B-ALL (Figure 2—figure supplement 3). We also show by ChIP-qPCR that PU.1 occupies the *Bim*-117 and Atp1b3-35 regions in 38B9, much like it does in B-ALL (Figure 3—figure supplement 2). We have also performed experiments in which we knock down *Bim* and PU.1 in 38B9, which we shows alleviates the TRIM33 requirement, in agreement with the observation in B-ALL (Figure 3—figure supplement 4 and Figure 4—figure supplement 1). Finally, we have also performed a CRISPR-based deletion of the *Bim*-117 element in 38B9, and found that this bypasses the TRIM33 requirement in this cell type (Figure 4—figure supplement 1).

Collectively, these findings provide strong support that the PU.1/TRIM33/*BIM* pathway is relevant in both 38B9 and B-ALL cell lines. In addition, we now have two independent cell lines in which we can show that TRIM33 performs an essential function through a single *cis* element.

*Minor points*:

*1) In the Discussion the authors claim TRIM33 as a therapeutic target “… since the major phenotypic abnormality in mice experiencing systemic Trim33 inhibition is a selective and reversible B lymphocyte deficiency.” The authors do not address the other hematopoietic lineage phenotypes previously described by Aucagne et al., J. Clin. Invest., 2011 and Bai et al., Dev. Biol., 2013, most importantly the CMML phenotype. That is surprising as the authors also see an expansion of the granulocytic compartment after 1 week (BM,*
Figure 1*), and more dramatically 4 weeks (spleen,*
Figure 1*). Please discuss this*.

In our revised manuscript, we now discuss the potential for tumorigenic effects incurred by TRIM33 inhibition in certain tissues, in light of the Aucagne et al. study showing CMML-like disease in TRIM33-ko mice. This appears in the last paragraph of the Discussion section.

*2) If PU.1 is a lineage-specific factor for many myeloid lineages, why would TRIM33 antagonize PU.1 specifically in B (progenitor) cells? What about PU.1 density at the* Bim *enhancers in AML or T-ALL cell lines (the authors only show TRIM33 occupancy in those)? One would expect that either PU.1 density is low or that* Bim *does not play a key role in these cell lines. This should be discussed*.

This is an important issue, which was not adequately addressed or discussed in the original manuscript. To carefully define the cell type-specific pattern of TRIM33 occupancy in these different leukemia contexts, we have now performed ChIP-seq analysis of TRIM33 in 38B9, AML, and T-ALL cell lines. This analysis shows that TRIM33 occupies the *Bim*-117 and *Atp1b3*-35 enhancers in B-ALL, 38B9, and AML, but not in T-ALL (Figure 2—figure supplement 1). Furthermore, we show that occupancy of TRIM33 at *Bim* and *Atp1b3* occurs at extremely high levels in 38B9 and AML as compared to other genomic sites, which resembles the skewed pattern observed in B-ALL (Figure 2—figure supplement 3). We have also performed ChIP-qPCR analysis of PU.1 in 38B9, AML, and T-ALL cell lines (Figure 2—figure supplement 2). This reveals PU.1 occupying *Bim*-117 and Atp1b3-35 in 38B9 and AML, but not T-ALL. Hence, these findings reveal that PU.1 and TRIM33 occupy similar elements at the *Bim* and Atp1b3 loci in B-lymphoid and myeloid leukemias, but not in T-lymphoid cells.

The ChIP-seq observation of TRIM33 occupancy at the *Bim*-117 enhancer in AML was unexpected, as our original manuscript included ChIP-qPCR data showing a lack of TRIM33 occupancy at this region in AML. This discrepancy prompted us to perform additional TRIM33 and PU.1 ChIP-qPCR experiments in 3 additional early passage AML cultures, which were all derived independently from the MLL-AF9/Nras^G12D^ mouse model. The results are shown below and validate the ChIP-seq findings that TRIM33 occupies the *Bim*-117 region in AML (Figure 5). We do not fully understand why our original results were erroneous with regard to TRIM33 occupancy in AML, but this may due to the use of a late-passage of AML cell lines for this experiment in the original submission, whereas the ChIP-seq and new ChIP-qPCR experiments was performed in early passage AML cultures. Alternatively, it is possible there was cell culture mixup that led to this incorrect conclusion. We apologize for this error in the original submission, which we have corrected in the revised manuscript.

These findings now raise the issue as to why *Bim* expression and apoptosis are unaffected in AML cells upon TRIM33 knockdown, despite PU.1/TRIM33 occupancy at *Bim* -117 in this cell type. An analysis of H3K27ac in B-ALL and AML reveals a significant difference in the *Bim* enhancer landscape in these two cell types. In B-ALL, the -117 kb region is among the most hyperacetylated regions in the vicinity of *Bim*, suggesting this enhancer is among the most active at the *Bim* locus in this cell type. In AML we observe a unique enhancer cluster at ∼66 kb upstream of *Bim* which is not observed in B-ALL (Figure 4—figure supplement 2). Notably, the levels of H3K27ac at this region are substantially higher than the level of H3K27ac at the -117 kb region. This raises the possibility that a different set of enhancers control *Bim* in AML, whose strength might be substantially greater than the activity of the -117 region. This leads us to speculate that, while TRIM33 and PU1 occupy the *Bim* locus in AML, this element is not a major contributor to *Bim* expression in this cell type, which is instead regulated by the -66 enhancer cluster. We provide a new Figure 4—figure supplement 2 and a new paragraph discussing this point in the revised manuscript (last paragraph of the Results section).

Author response image 1.ChIP-qPCR of TRIM33 and PU.1 in 3 independent MLL-AF9/Nras^G12D^ acute myeloid leukemia cell lines, which are all early passage cultures derived from leukemic mice.**DOI:**
http://dx.doi.org/10.7554/eLife.06377.023

*3) How do the authors imagine the interplay of TRIM33 and PU.1 in normal B cell differentiation? Is this mechanism active in vivo? The authors show that in the mouse model depleted for Trim33 B-lymphoid cells are selectively lost. They do not address whether apoptosis in the model is due to PU.1-TRIM33 antagonism. In fact, they show that TRIM33 binds to the* Bim *enhancer in normal spleen. However, work by Oliver et al., J. Exp. Med., 2004 shows that* Bim *is expressed in B-cell precursors. Is this also true in the authors’ system? If so, how do they explain* Bim *expression despite TRIM33 occupancy at the enhancer? What are the absolute* Bim *levels compared to those in the B-ALL model?*

Our manuscript does not completely address whether the TRIM33/PU.1 antagonism is operating during normal B cell development. However, there are several lines of suggestive evidence that point to the observations in B-ALL being relevant during normal B cell differentiation. First, we observe a similar sensitivity in normal and transformed B cells to TRIM33 knockdown. Second, we observe PU.1 binding to the *Bim*-117 and *Atp1b3*-35 regions in previously published ChIP-seq datasets obtained from normal B cell progenitors (Heinz and Glass, Molecular Cell 2010, PMID: 20513432). Third, we observe TRIM33 occupancy at the *Bim*-117 and *Atp1b3*-35 regions in whole spleen. All of these findings would be consistent with TRIM33/PU.1 antagonism being relevant to the phenotype observed in TRIM33-deficient mice, however definitive evidence will require additional genetic experiments in normal B lymphoid cells. We describe the potential relevance of TRIM33 for regulating normal B cell development in the Discussion section.

It should be noted that *Bim* is actually expressed at reasonably high levels in B-ALL cells even in the presence of TRIM33 (RPKM value of 12.6 from B-ALL RNA-seq). It can also be appreciated in Figure 2 that the *Bim* promoter exhibits robust H3K4me3 in B-ALL despite TRIM33 occupancy at the -117 region. In addition, by Western blotting we can readily detect *Bim* protein in TRIM33-proficient B-ALL cells (Figure 4). The *Bim* locus in B-ALL cells also harbors multiple H3K27ac peaks, suggesting multiple active enhancers are contributing to *Bim* expression, with TRIM33 only antagonizing one of these elements. Hence, our findings are consistent with Oliver et al., 2004, in that *Bim* is expressed in these different normal and malignant B cells, but this expression is just below the critical threshold for committing cells to undergo apoptosis. Hence, targeting TRIM33 tips this balance by quantitatively increasing *Bim* expression through releasing the activity of one enhancer. It does not appear that TRIM33 regulates an on/off switch of *Bim* expression. For space considerations of a short report, we have not provided an extensive description of this issue in the text of the manuscript.

*4) The fact the GFP-gated cells in vivo show more efficient B-lineage depletion than non-GFP-gated cells (and that there are non-GFP cells present under induced conditions in the marrow) suggests to me that the mouse strain with inducible shRNA/GFP has variegated expression in the B-linage or even whole BM (position effect). I believe this does not affect the manuscript's message, but the authors should clarify*.

In the subsection headed “Transgenic animal studies” of the Methods section, we describe in greater detail the GFP percentages in the bone marrow and spleen following dox administration and the variegated expression pattern of shRNA/GFP in these animals. The bone marrow is approximately 75% GFP+ and the spleen is approximately 15% GFP+, which highlights the need for GFP gating to accurately measure effects on normal hematopoietic cells.

*5) The authors allude to other factors with documented peak number: MyoD ∼39,000 and Pax5 ∼20,000 sites (*[9]*;*
[33]*). In a side by side comparison, using the same peakfinder and same parameters including p-value cutoff, how many significant peaks are there for Trim33 in the p190-leukemia cells? It might be that not the number of peaks is different from characterized transcriptional regulators but the existence of the unusual outliers including* Bim*. Please clarify*.

The Pax5 study used the same algorithm (MACS) as we have used to characterize TRIM33 occupancy. In fact, in the Pax5 ChIP-seq analysis the authors used an even greater stringency of cutoffs than we have used when we defined 31 TRIM33 peaks in B-ALL. Hence, applying the same cutoffs to the TRIM33 ChIP-seq data would have revealed even fewer peaks than the 31 we were able to identify. As described below, the absolute number of peaks identified using ChIP-seq must caveated by the arbitrary enrichment cutoffs used to define occupancy.

Our revised manuscript now includes TRIM33 ChIP-seq performed in 38B9, AML, and T-ALL. This analysis reveals a substantially greater number of peaks overall in each of these cell types that achieves statistical significance using standard MACS peak calling criteria. However, we note in each dataset that there are indeed, as the reviewers suggest, the presence of unusual outliers with exceptional occupancy, such as *Bim* and Atp1b3. Throughout the text of the revised manuscript, we have more clearly described this type of pattern. For instance, we have removed the description of TRIM33 occupancy as ‘sparse’ in the B-ALL genome (which was the wording in the original Abstract), since we should not over-interpret the absolute number of peaks identified in the ChIP-seq experiments. Instead, it is more appropriate to describe the disproportionate accumulation of TRIM33 at a small number of sites in each cell type, which appears to be a consistent pattern in B-ALL, 38B9, and AML, but to a lesser extent in T-ALL. Our revised manuscript also includes a MEME-based motif analysis of the TRIM33 peaks in each of these cell types, which confirms the relevance of PU.1 in myeloid and B lymphoid cell types, but reveals additional TFs as contributing the TRIM33 recruitment in T-ALL (Figure 2—figure supplement 3).